# VISUAL PROGRAMMABILITY: A GUIDE FOR CODE-AS-THOUGHT IN CHART UNDERSTANDING

## ABSTRACT

Chart understanding presents a critical test to the reasoning capabilities of Vision-Language Models (VLMs). Prior approaches face critical limitations: some rely on external tools, making them brittle and constrained by a predefined toolkit, while others fine-tune specialist models that often adopt a single reasoning strategy, such as text-based chain-of-thought (CoT). The intermediate steps of text-based reasoning are difficult to verify, which complicates the use of reinforcement-learning signals that reward factual accuracy. To address this, we propose a Code-as-Thought (CaT) approach to represent the visual information of a chart in a verifiable, symbolic format. Our key insight is that this strategy must be adaptive: a fixed, code-only implementation consistently fails on complex charts where symbolic representation is unsuitable. This finding leads us to introduce **Visual Programmability**: a learnable property that determines if a chart-question pair is better solved with code or direct visual analysis. We implement this concept in an **adaptive** framework where a VLM learns to choose between the CaT pathway and a direct visual reasoning pathway. The selection policy of the model is trained with reinforcement learning using a novel dual-reward system. This system combines a data-accuracy reward to ground the model in facts and prevent numerical hallucination, with a decision reward that teaches the model when to use each strategy, preventing it from defaulting to a single reasoning mode. Experiments demonstrate strong and robust performance across diverse chart-understanding benchmarks. Our work shows that VLMs can be taught not only *to* reason but also *how* to reason, dynamically selecting the optimal reasoning pathway for each task.

## 1 INTRODUCTION

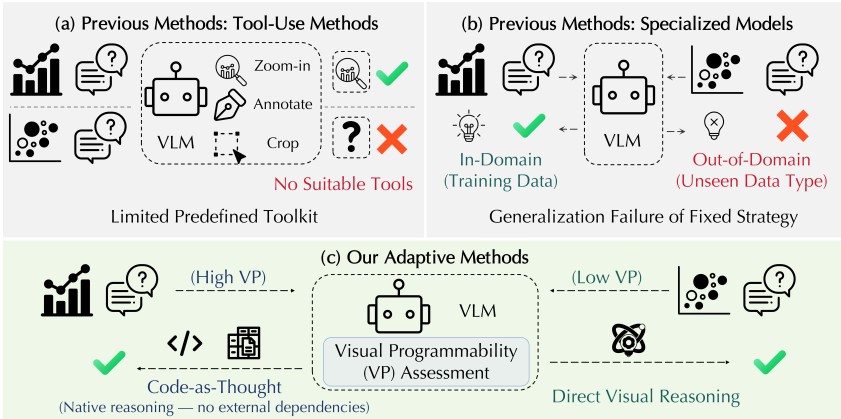

**Figure 1:** Adaptive Reasoning vs. Fixed Strategies for Chart Understanding. Prevailing approaches are limited by their rigid strategies. (a) Tool-Use Models are constrained by a predefined toolkit and fail on novel tasks. (b) Specialized Models employ a single reasoning pattern (e.g., text-only or code-only), which limits their generalization. In contrast, our (c) Adaptive Framework first assesses a task's Visual Programmability. It then dynamically selects the precise Code-as-Thought pathway for programmable tasks or the robust **Direct Visual Reasoning** pathway for complex ones, achieving superior performance across all chart types.

The capabilities of Vision-Language Models (VLMs), built upon powerful Large Language Models Brown et al. (2020); Touvron et al. (2023), have rapidly advanced multimodal understanding (e.g., Radford et al. (2021); Liu et al. (2023); Achiam et al. (2023); Comanici et al. (2025); Bai et al. (2025)). Among the many applications, chart understanding stands out as a critical benchmark Huang et al. (2024), testing an AI's ability to connect low-level visual perception Lee et al. (2023) with high-level logical inference. Despite significant progress with specialized models Cheng et al. (2023); Masry et al. (2023); Meng et al. (2024), a fundamental generalization problem remains: even state-of-the-art VLMs show a stark performance decline on the complex, "in-the-wild" charts found in real-world contexts Islam et al. (2024); Wang et al. (2024).

Prevailing efforts to overcome this generalization challenge have largely followed two dominant strategies, each with distinct drawbacks. The first approach treats the VLM as a controller for external tools and APIs Huang et al. (2025); Gupta & Kembhavi (2023); Surís et al. (2023) (see Figure 1a). While powerful, their reliance on a predefined toolkit makes them brittle when encountering charts that require capabilities beyond their predefined functions Schick et al. (2023); Yao et al. (2023b); Patil et al. (2024); Parisi et al. (2022). The second strategy involves fine-tuning specialized models on chart-specific data Cheng et al. (2023); Masry et al. (2023); Meng et al. (2024) (see Figure 1b). These models typically rely on a *monolithic reasoning pattern*—that is, they exclusively use a single mode of thought, such as text-based Chain-of-Thought or code-based reasoning. This lack of flexibility hinders their ability to generalize to out-of-distribution (OOD) visualizations, as no single reasoning style is optimal for all chart types Wang et al. (2024); Xu et al. (2023).

The limitations of predefined toolkits highlight the appeal of a more universal and flexible tool: code. Unlike a fixed API, code can be dynamically generated to create novel tools tailored to the specific visual complexities of any chart, a concept explored in recent agentic vision systems Zhao et al. (2025a). However, the shared failure of rigid approaches motivates our core insight: the optimal reasoning strategy depends on the task itself. Some charts are easily broken down into programmable elements Dai et al. (2024), while others require a holistic visual analysis that code cannot capture. This requires moving beyond refining a single reasoning chain Wei et al. (2022) to mastering strategy selection—a shift that reflects a broader trend in AI towards deliberate problem-solving Wang et al. (2022); Shinn et al. (2023); Yao et al. (2023a) and adaptive computation Graves (2016). This principle is also central to the design of frontier models like GPT-5 OpenAI (2025), which aim to integrate similar adaptive capabilities.

To address these challenges, we propose the concept of **Visual Programmability**: a learnable, task-dependent property that indicates whether a given chart-question pair is best solved through programmatic reasoning or direct visual analysis. We implement this concept in an adaptive framework that enables a VLM to autonomously choose its reasoning pathway. The model's decision-making policy is trained via reinforcement learning (RL)—specifically, using the Group Relative Policy Optimization (GRPO) algorithm—guided by a novel **dual-reward system**. This system is carefully designed to foster adaptive behavior: a data-accuracy reward ensures the generated code is factually grounded to the chart's content, thereby preventing numerical hallucination. In parallel, a dedicated decision reward explicitly teaches the model the boundaries of programmability, preventing the policy from collapsing into a single, suboptimal mode.

Experiments with Qwen2.5-VL Bai et al. (2025) across diverse benchmarks validate our approach. The adaptive model outperforms pure-vision and fixed code baselines by selecting its strategy—using code when advantageous (**>60%**) and avoiding it when harmful (<10%). Ablations confirm that the dual-reward design prevents mode collapse and promotes strategic diversity. Our contributions are threefold:

- We introduce **Visual Programmability**, a novel concept to determine if a chart task is suitable for code-based reasoning, serving as the foundation for adaptive strategy selection.

- Building on this concept, we develop an **adaptive framework** that learns to choose the optimal reasoning path (code or vision). This framework is trained with a specialized dual-reward RL system that promotes both factual accuracy and strategic flexibility.

- Our adaptive model demonstrates **outstanding performance and generalization**, consistently outperforming rigid strategies across diverse benchmarks by intelligently switching between reasoning modes.

## 2 RELATED WORK

**Programmatic Reasoning and Its Limits.** The field of chart understanding has seen a shift towards programmatic reasoning. This includes VLMs acting as controllers for external tools Gao et al. (2023); Schick et al. (2023); Surís et al. (2023) and models that generate code as a form of symbolic thought Subramanian et al. (2023); Chen et al. (2023). While specialized models achieved high scores on earlier benchmarks Cheng et al. (2023); Masry et al. (2023), their success was often misleading. They tended to learn benchmark-specific shortcuts, a weakness exposed by a new wave of diverse and complex benchmarks Xia et al. (2024); Wang et al. (2024); Xu et al. (2023), where even state-of-the-art models show a significant performance drop Islam et al. (2024); Huang et al. (2024). We argue that this generalization gap stems not from a lack of model capability, but from *strategic rigidity*. Despite variations in approach—from Mixture-of-Experts Xu et al. (2024) to visual grounding techniques Ni et al. (2025); Huang et al. (2025)—existing methods adhere to a fixed reasoning pattern. Our work departs from this by reframing the challenge: instead of augmenting VLMs with external tools, we teach them to recognize *when* to deploy their own code-like reasoning, shifting the focus from tool use to strategic selection.

**Adaptive Learning for Strategic Reasoning.** Our framework builds on the idea of adaptive computation, where a system alters its strategy based on the input Bengio et al. (2015); Kahneman (2011). In AI, this is often implemented through methods like dynamic routing or Mixture-of-Experts layers, which adapt *how* a model performs its computation Sabour et al. (2017); Jiang et al. (2024). We apply this concept at a higher level: we teach a model to decide *what* reasoning process to use, a skill we call *strategic cognition*. Reinforcement learning (RL) is well-suited for learning such a policy from outcome-based feedback, a technique proven effective for tasks with verifiable answers Meng et al. (2025); Su et al. (2025); Lightman et al. (2023). However, a simple accuracy reward can cause the model to always default to a single, safe strategy—a problem known as *mode collapse*. Our key contribution is a **dual-reward system** that combines an accuracy signal with a dedicated decision reward. This design encourages strategic diversity, teaching the model not just to solve the task, but how to choose the right reasoning tool for the job.

## 3 EXPLORING CODE-AS-THOUGHT AS A UNIVERSAL STRATEGY

The limitations of the fixed strategies discussed previously motivate us to explore whether a more powerful, formal paradigm could serve as a universal solution for chart understanding. This line of inquiry leads us to investigate Code-as-Thought (CaT) and to pose a foundational question:

*Is Code-as-Thought a "silver bullet" for chart understanding?*

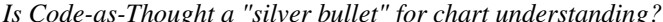
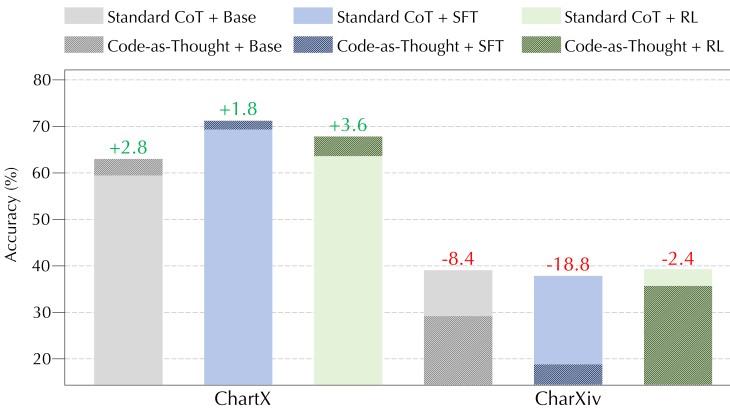

Figure 2: **Performance of Fixed Strategies Highlights a Critical Trade-off.** While the Code-as-Thought (CaT) strategy excels on structured charts (ChartX), its performance collapses on complex, 'in-the-wild' charts (CharXiv). All values are accuracy (%).

To answer this, we first investigated the efficacy of a single, fixed CaT strategy. We trained a specialist model on structured data and evaluated its generalization across four diverse benchmarks.

We discovered a core limitation that motivates our adaptive framework. Figure 2 visualizes the results on two of these benchmarks—the highly structured ChartX and the complex, "in-the-wild" CharXiv—which most clearly illustrate the performance trade-offs. A detailed description of the setup and full results across all four benchmarks are provided in Appendix A.

The results reveal a sharp dichotomy in generalization performance. As shown in Figure 2, the CaT specialist (achieving 71.6% with SFT) excels on the structured ChartX data, confirming its power in high-programmability scenarios. However, this rigid strategy proves brittle. On the complex charts from CharXiv, its accuracy collapses to a mere 18.4%. This failure is often driven by numerical hallucination—where the model generates code from a flawed perception of the chart, then reasons faithfully from this incorrect foundation. A case of this phenomenon is detailed in Appendix B.

Furthermore, we found that enhanced skill and policy optimization are not a panacea. The right side of the figure illustrates that even after applying reinforcement learning (RL), the model's performance on CharXiv remains critically low, failing to resolve the core conflict. Results with extensive pre-training (CPT+RL) exhibit the same trend and are provided in Appendix A. The conclusion is clear: the issue is not the model's competence (how well it codes) but determining the strategy's applicability (whether it *should* code at all). These experiments confirm the potential of Code-as-Thought but reveal that the optimal strategy is task-dependent, motivating our core thesis: an intelligent system must learn *when* to use its tools, not just how.

## 4 ADAPTIVE CODE-BASED REASONING FRAMEWORK

Our framework enables a Vision-Language Model (VLM) to dynamically select the optimal reasoning strategy for a given chart. As illustrated in Figure 3, it consists of three core parts: an adaptive inference system, a training process based on reinforcement learning, and the underlying concept of Visual Programmability that guides the model's learning.

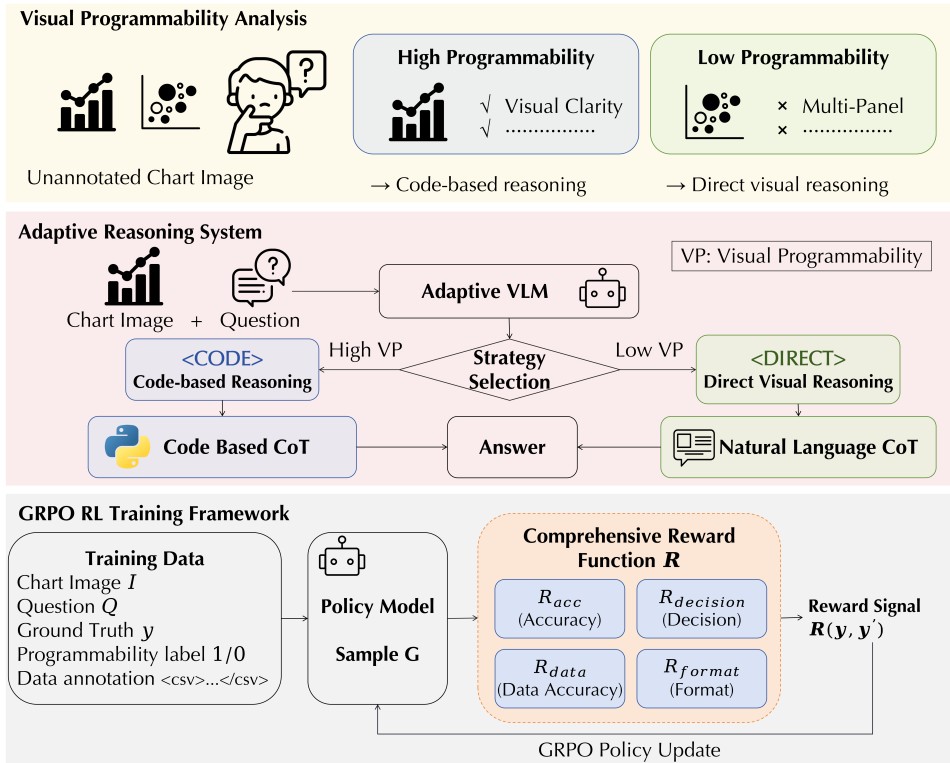

**Figure 3:** Overview of our adaptive reasoning framework. **(Top)** We introduce the concept of Visual Programmability and use it to guide data annotation. **(Middle)** At inference, our adaptive VLM selects a reasoning pathway based on the perceived Visual Programmability (VP) of the task. **(Bottom)** The model's selection policy is trained using reinforcement learning with a multi-component reward function and the GRPO algorithm.

## 4.1 VISUAL PROGRAMMABILITY: UNDERSTANDING THE BOUNDARIES OF CODE

Not all charts are equally well-suited to analysis using Code-as-Thought. To address this, we introduce the concept of **Visual Programmability**: a learnable, task-dependent property that serves as the foundation for our adaptive reasoning system. It determines whether a chart-question pair can be faithfully reasoned using code. This property is not a binary yes-or-no question; rather, it represents a range of suitability influenced by a chart's structural clarity, its visual complexity, and the query itself. Figure 4 provides several cases that illustrate this concept.

**High vs. Low Programmability.** The suitability of code-based reasoning varies widely. Some charts exhibit high programmability. These are typically standard bar, line, or scatter plots with clean layouts, where the underlying data can be programmatically extracted with high fidelity. Figure 4 (a) shows a clear example: a standard line chart with explicitly marked data points, making it ideal for precise computational analysis. In contrast, other charts have low programmability. As seen in Figure 4 (b), these often include complex scientific visualizations where meaning is conveyed through holistic patterns, such as data contours and distributions. For these charts, essential information is often lost or distorted during symbolic translation.

**The Critical Role of Task Dependency.** Crucially, Visual Programmability is not an intrinsic chart property alone; it is fundamentally dependent on the user's query. This is demonstrated by the case in Figure 4 (c). For a simple counting task like, *"How many distinct data series are plotted?"*, the chart has **high programmability**, as the task only requires identifying discrete visual elements. However, for a value-extraction task like, *"What is the approximate value of the orange line (h/a = 1000) when d = 7?"*, the same chart exhibits **low programmability**. The logarithmic scale makes precise data extraction extremely difficult and error-prone. In this scenario, a Code-as-Thought approach would likely yield a confidently incorrect answer, making direct visual reasoning a more reliable strategy.

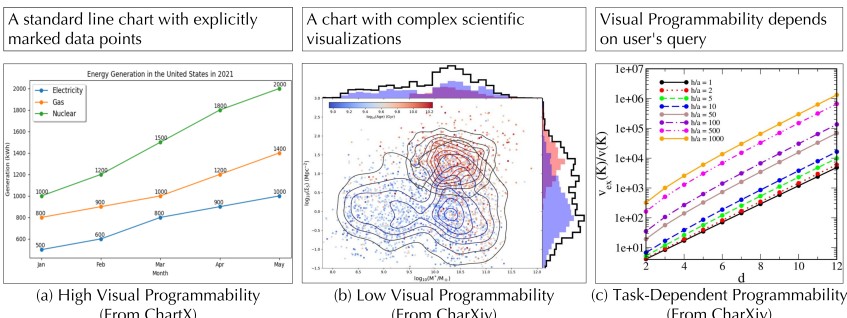

| A standard line chart with explicitly marked data points | A chart with complex scientific visualizations | Visual Programmability depends on user's query |
|---|---|---|

(a) High Visual Programmability (From ChartX)  (b) Low Visual Programmability (From CharXiv)  (c) Task-Dependent Programmability (From CharXiv)

**Figure 4:** Cases of Visual Programmability for different charts and tasks.

This dependency on both the chart and the question necessitates a dynamic reasoning system. An intelligent agent cannot rely on a fixed strategy; it must learn to assess Visual Programmability to select the most appropriate reasoning path. To enable this, we developed a framework to annotate data for this property, providing the signal for learning this adaptive skill (see Appendix C).

## 4.2 ADAPTIVE REASONING MECHANISM

We formulate the chart-understanding task as a policy learning problem. Given a chart image $\mathbf{I}$ and a question $\mathbf{Q}$, our model learns a policy $\pi_\theta$ that generates a response $\mathbf{y}$. This process is explicitly factorized to first select a strategy token $s \in \{\texttt{}, \texttt{<DIRECT>}\}$, then generate the corresponding reasoning and answer:

$$P(\mathbf{y}|\mathbf{I}, \mathbf{Q}) = P(s|\mathbf{I}, \mathbf{Q}) \cdot P(\mathbf{y}|\mathbf{I}, \mathbf{Q}, s). \tag{1}$$

This factorization is realized by building upon powerful base models (Qwen2.5-VL) and teaching them to first commit to a strategy by generating a special token, which then dictates the subsequent generation path:

**Code-based Path (``):** The model generates a Code-as-Thought (CaT) pathway. It writes code to parse the chart into a structured format (e.g., a DataFrame) and then performs computations to find the answer. This path is ideal for charts with high Visual Programmability.

**Direct Path (`<DIRECT>`):** The model generates a natural language CoT, performing reasoning based on its holistic visual perception. This path is essential for charts with low Visual Programmability where symbolic decomposition would lose critical information.

For automated evaluation, the final answer from both paths must be enclosed in `\boxed{}`.

### 4.3 TRAINING VIA REINFORCEMENT LEARNING

The crucial challenge is the absence of ground-truth labels for strategy selection. We overcome this by formulating the training as a reinforcement learning problem, allowing the model to learn the optimal policy from outcome-based reward signals.

#### 4.3.1 GRPO POLICY UPDATE

We optimize the policy with Group Relative Policy Optimization (GRPO). For each input, we sample a group of $G$ rollouts from $\pi_{\text{old}}$, score them with our reward, compute group-normalized advantages, and apply a clipped-ratio update. We follow a KL-free configuration by setting $\beta=0$. The full objective, notation, and update details are provided in Appendix G.

#### 4.3.2 COMPREHENSIVE REWARD FUNCTION

A naive reward function focused solely on answer accuracy would be insufficient and could lead to *mode collapse*—where the model defaults to a single, suboptimal strategy. To prevent this and guide the model toward true adaptive behavior, we designed a comprehensive reward function $R$ as a weighted sum of four specialized components:

$$R = w_{\text{acc}}r_{\text{acc}} + w_{\text{decision}}r_{\text{decision}} + w_{\text{data}}r_{\text{data}} + w_{\text{format}}r_{\text{format}}. \tag{2}$$

The components are:

**Accuracy Reward ($r_{\text{acc}}$):** The primary reward, providing a binary signal (1.0 or 0.0) based on the correctness of the final answer.

**Decision Reward ($r_{\text{decision}}$):** Our key innovation to prevent mode collapse. This reward explicitly incentivizes selecting the correct strategy based on the chart's pre-annotated Visual Programmability. It gives a full reward for a correct answer via the correct strategy, a partial reward for a wrong answer but using the correct strategy (to encourage exploration), and zero reward for using the wrong strategy. This component is essential for teaching the model to learn the decision boundary.

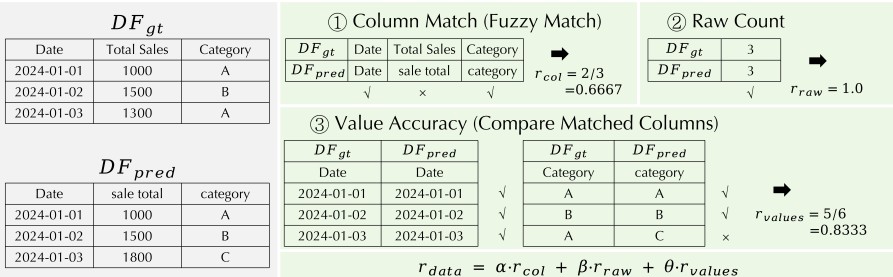

**Figure 5:** Illustration of the Data Accuracy Reward calculation.

**Data Accuracy Reward ($r_{\text{data}}$):** Applied *only* to the `` path, this reward tackles the issue of code "hallucination." It programmatically compares the DataFrame generated by the model's code to a ground-truth data table, evaluating the fidelity of the extracted data. This ensures the model generates code that is not just syntactically valid, but semantically faithful to the chart. The calculation process is visualized in Figure 5.

**Format Reward ($r_{\text{format}}$):** A small reward to enforce correct output structure (i.e., using `\boxed{}`), ensuring reliable parsing.

This multi-faceted reward design creates a nuanced optimization landscape that simultaneously pushes the model toward accuracy and strategic intelligence. The detailed implementation of the Data Accuracy Reward is provided in Appendix E.

## 5 EXPERIMENTS

### 5.1 EXPERIMENTAL SETUP

We train on ChartMimic Yang et al. (2024) augmented with automatically generated QA pairs (Gemini-2.5-Flash; prompts in Appendix F) to cover both programmable and non-programmable chart–question pairs. We evaluate on four benchmarks spanning the Visual Programmability spectrum: ChartX Xia et al. (2024), ChartBench (NQA) Xu et al. (2023), ChartQA Masry et al. (2022), and CharXiv Wang et al. (2024). Our base is Qwen2.5-VL-7B trained with GRPO (EasyR1 Zheng et al. (2025)) under the dual-reward design in Eq. 2. Full dataset composition, sampling protocol, metric definitions, and hyperparameters are detailed in Appendix D.

### 5.2 COMPARISON WITH FIXED-STRATEGY BASELINES

**Table 1:** Comparison with fixed-strategy baselines on four benchmarks. Our adaptive RL model achieves the highest average accuracy by dynamically selecting its reasoning strategy. All values are accuracy (%).

| Model Type | Reasoning Strategy | ChartX | ChartBench | ChartQA | CharXiv | Average |
|---|---|---|---|---|---|---|
| Base Models (No RL) | Standard CoT | 59.2 | 50.1 | 84.9 | 38.4 | 58.2 |
| | Code CoT (Fixed) | 59.8 | 53.4 | 79.4 | 28.8 | 55.4 |
| | Adaptive | 57.8 | 51.4 | 84.4 | 22.8 | 54.1 |
| RL Models | Standard CoT | 61.5 | 52.8 | 86.6 | 43.8 | 61.2 |
| | Code CoT (Fixed) | 64.0 | 54.0 | **86.7** | 41.9 | 61.7 |
| | **Adaptive (Ours)** | **65.6** | **54.8** | 86.4 | **44.3** | **62.8** |

As shown in Table 1, our adaptive framework achieves the highest average accuracy (**62.8%**), outperforming all fixed-strategy baselines. This advantage stems from its learned ability to dynamically select the optimal reasoning path.

Table 2 reveals this strategic behavior. On high-programmability benchmarks like ChartX and ChartBench, our model favors the code-based path (**76.0%** and **66.6%** usage) to leverage its precision. On the complex CharXiv benchmark, it astutely reduces code usage to just **10.1%**, avoiding the pitfalls of a rigid code-only approach and achieving the highest accuracy (**44.3%**). The results on ChartQA further suggest that our Data Accuracy Reward improves not only *when* the model uses code, but also *how reliably* it does so.

Qualitative examples (see Appendix I) further highlight this strategic intelligence: our model correctly selects the  path for precise calculations on structured charts and the <DIRECT> path for complex plots, successfully navigating scenarios where fixed-strategy baselines fail.

**Table 2:** Code usage percentage across benchmarks for our adaptive model versus fixed strategies. The model learns to apply code frequently on high-programmability charts and sparingly on low-programmability ones. All values are percentages (%).

| Model Type | Reasoning Strategy | ChartX | ChartBench | ChartQA | CharXiv |
|---|---|---|---|---|---|
| Base Models (No RL) | Standard CoT | 0.0 | 0.0 | 0.0 | 0.0 |
| | Code CoT (Fixed) | 98.9 | 100.0 | 98.3 | 99.5 |
| | Adaptive | 99.7 | 99.6 | 98.8 | 92.9 |
| RL Models | Standard CoT | 0.0 | 0.0 | 0.0 | 0.0 |
| | Code CoT (Fixed) | 100.0 | 100.0 | 100.0 | 100.0 |
| | **Adaptive (Ours)** | **76.0** | **66.6** | **98.3** | **10.1** |

### 5.3 COMPARISON WITH STATE-OF-THE-ART MODELS

To contextualize our results, we compare our adaptive framework against several state-of-the-art (SOTA) models. All models, unless noted, were evaluated under our stringent protocol to ensure a fair comparison. As shown in Table 3, our model achieves the highest average accuracy (**62.8%**), significantly outperforming other SOTA models. This performance gap, especially on diverse benchmarks like ChartX and CharXiv, underscores the advantage of our adaptive reasoning approach.

**Table 3:** Comparison with state-of-the-art models on four key generalization benchmarks. Our model demonstrates outstanding performance, achieving the highest average accuracy. All values are percentages (%).

| Model | Parameters | ChartX | ChartBench | ChartQA | CharXiv | Average |
|---|---|---|---|---|---|---|
| ChartVLM-Large Xia et al. (2024) | 8.3B | 35.0 | 28.8 | 66.7 | 14.7 | 36.3 |
| ChartGemma Masry et al. (2024) | 3B | 28.7 | 27.5 | 69.0 | 20.3 | 36.4 |
| ChartMoE Xu et al. (2024) | 8B | 33.6 | 29.5 | 74.2 | 28.3 | 41.4 |
| Orsta-7B Ma et al. (2025) | 7B | 60.3 | 52.0 | 84.6 | 41.5 | 59.6 |
| Point-RFT Ni et al. (2025) | 7B | - | - | $90.04^{\dagger}$ | 36.02* | - |
| Thyme-VL Zhang et al. (2025) | 7B | - | - | 86.1* | - | - |
| **Ours (Adaptive)** | **7B** | **65.6** | **54.8** | 86.4 | **44.3** | **62.8** |

*Results are taken directly from the original paper.
$^{\dagger}$In-domain evaluation result taken from the original paper.

## 5.4 ANALYSIS ON DIFFERENT MODEL SCALES

Our approach demonstrates strong scalability. On the 32B model (Table 4), our adaptive framework achieves the highest average accuracy (61.0%) and top performance on the challenging ChartX and CharXiv benchmarks. The results from the 3B model are more nuanced. While the fixed 'Code CoT' strategy yields the best average performance (56.5%), we hypothesize the adaptive strategy is constrained by the smaller model. It is nonetheless striking that after RL, the 'Standard CoT' model's performance collapses (from 31.9% to 20.4%), whereas both code-based strategies improve substantially. This strongly indicates that our structured, code-centric reward system provides a more stable and effective learning signal than a simple accuracy reward on free-form text.

**Table 4:** Performance comparison on 3B and 32B models. Our adaptive framework scales effectively to larger models, achieving the best overall performance on the 32B scale. The best results in each RL-trained category are highlighted in **bold**. All values are accuracy (%).

| Model Size | Training | Reasoning Strategy | ChartX | ChartBench | ChartQA | CharXiv | Average |
|---|---|---|---|---|---|---|---|
| 3B | Base Model (No RL) | Standard CoT | 48.0 | 39.2 | 13.8 | 26.7 | 31.9 |
| | | Code CoT (Fixed) | 51.3 | 42.0 | 28.0 | 29.3 | 37.7 |
| | | Adaptive | 1.0 | 0.7 | 0.3 | 10.6 | 3.2 |
| | RL-Trained | Standard CoT | 9.3 | 9.3 | 41.8 | 21.3 | 20.4 |
| | | Code CoT (Fixed) | **58.5** | **48.5** | **82.3** | **36.7** | **56.5** |
| | | Adaptive (Ours) | 55.6 | 43.5 | 73.6 | 33.6 | 51.6 |
| 32B | Base Model (No RL) | Standard CoT | 53.7 | 47.2 | 83.4 | 36.3 | 55.2 |
| | | Code CoT (Fixed) | 56.3 | 49.6 | 84.8 | 39.9 | 57.7 |
| | | Adaptive | 56.6 | 45.7 | 84.4 | 37.7 | 56.1 |
| | RL-Trained | Standard CoT | 54.7 | 47.9 | 84.6 | 35.9 | 55.8 |
| | | Code CoT (Fixed) | 59.6 | **49.5** | **87.9** | 44.5 | 60.4 |
| | | **Adaptive (Ours)** | **60.2** | 48.4 | 87.7 | **47.5** | **61.0** |

## 5.5 ABLATION STUDIES

### 5.5.1 DISSECTING THE REWARD FUNCTION

Tables 5 and 6 show the effect of our reward components. The **Decision Reward** ($r_{\text{decision}}$) prevents mode collapse, without which the model defaults to a rigid 0/100% code usage. While $r_{\text{decision}}$ teaches *when* to use code, the **Data Accuracy Reward** ($r_{\text{data}}$) teaches *how* to use it well, preventing over-caution. Together, they create a balanced policy for optimal performance.

**Table 5:** Ablation study on reward components. The full reward function is essential for achieving the highest accuracy. All values are accuracy (%).

| Reward Configuration | ChartX | ChartBench | ChartQA | CharXiv | Average |
|---|---|---|---|---|---|
| $r_{\text{acc}} + r_{\text{format}}$ (Baseline) | 62.2 | 52.2 | **86.5** | 43.6 | 61.1 |
| + $r_{\text{data}}$ (w/o $r_{\text{decision}}$) | 64.3 | 53.5 | 86.4 | 39.4 | 60.9 |
| + $r_{\text{decision}}$ (w/o $r_{\text{data}}$) | 63.6 | 52.4 | 86.3 | 43.3 | 61.4 |
| **Full Reward (Ours)** | **65.6** | **54.8** | 86.4 | **44.3** | **62.8** |

**Table 6:** Code usage percentage in the reward ablation study. The decision reward ($r_{\text{decision}}$) is critical for preventing mode collapse and enabling adaptive behavior. All values are percentages (%).

| Reward Configuration | ChartX | ChartBench | ChartQA | CharXiv |
|---|---|---|---|---|
| $r_{\text{acc}} + r_{\text{format}}$ (Baseline) | 0.0 | 0.0 | 0.0 | 0.0 |
| $+ r_{\text{data}}$ (w/o $r_{\text{decision}}$) | 100.0 | 100.0 | 100.0 | 100.0 |
| $+ r_{\text{decision}}$ (w/o $r_{\text{data}}$) | 50.4 | 11.0 | 87.4 | 0.7 |
| **Full Reward (Ours)** | **76.0** | **66.6** | **98.3** | **10.1** |

### 5.5.2 THE CRITICAL ROLE OF NUMERICAL FIDELITY

This analysis confirms the importance of our data accuracy reward. As shown in Table 7, there is a direct and stark correlation between the fidelity of extracted data and the final answer accuracy. High-fidelity extraction leads to an impressive **85.6%** accuracy, demonstrating that correct data extraction is a prerequisite for successful reasoning on programmable charts. As Figure 6 illustrates, our data accuracy reward ($r_{\text{data}}$) grounds the model by teaching it to improve on high-fidelity tasks while "unlearning" to guess on low-fidelity ones.

**Table 7:** The stark correlation on the ChartX benchmark between the accuracy of extracted numerical data and final answer correctness. High-fidelity data extraction is demonstrably a prerequisite for success.

| Numerical Accuracy Score ($r_{\text{data}}$) | Final Answer Accuracy (%) |
|---|---|
| < 0.6 (Low Fidelity) | 48.4 |
| 0.6 - 0.8 (Medium Fidelity) | 60.5 |
| > 0.8 (High Fidelity) | **85.6** |

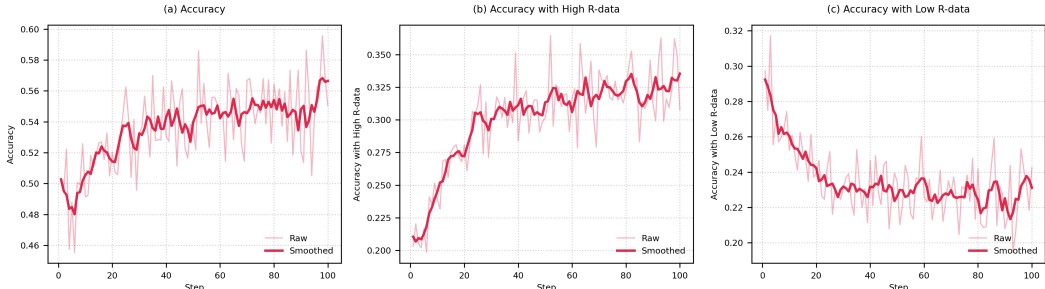

**Figure 6:** Training dynamics on ChartX, illustrating the effect of the Data Accuracy Reward ($r_{\text{data}}$). **(Left)** Overall task accuracy increases. **(Middle)** Accuracy with high data fidelity ($r_{\text{data}} > 0.6$) rises sharply. **(Right)** Accuracy with low data fidelity ($r_{\text{data}} < 0.6$) trends downward, as the model unlearns to guess.

## 6 CONCLUSION

We challenged the prevailing one-size-fits-all paradigm in visual reasoning. To this end, we introduced **Visual Programmability**, a concept that explains why powerful Code-as-Thought (CaT) strategies excel on structured charts but fail on complex ones. Building on this insight, we developed an adaptive framework trained with a novel dual-reward system. Our model learns to dynamically select between the precision of CaT and the robustness of direct visual reasoning, deploying the optimal strategy for each task. The key insight from our work is that robust, general-purpose reasoning emerges not from a superior monolithic strategy, but from the meta-cognitive skill of knowing one's own capabilities and limitations. This work provides a concrete blueprint for building more flexible AI systems—systems that don't just follow procedures, but strategically decide how to think. A detailed discussion of limitations, future work, and broader implications is provided in Appendix H.

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

## THE USE OF LARGE LANGUAGE MODELS

We used Gemini 2.5 Pro for the following limited purposes: (i) language polishing of paragraphs; (ii) generating boilerplate code for plotting; and (iii) drafting figure captions. All scientific claims, methods, and results were conceived, verified, and validated by the authors. We manually checked and reproduced any outputs suggested by the LLM. No confidential or identifying information was provided to the LLM service.

## A   DETAILED ANALYSIS OF FIXED-STRATEGY EXPERIMENTS

**Experimental Setting.**   To create our specialist model, we fine-tuned Qwen2.5-VL-7B using a Supervised Fine-Tuning (SFT) approach on the ChartX validation set Xia et al. (2024). This dataset consists of approximately 4,800 highly structured charts well-suited for programmatic analysis. We then evaluated this specialized model's generalization ability across four diverse test suites, each containing 500 samples designed to span a spectrum of difficulty and style:

- **In-Domain (ChartX Xia et al. (2024)):** A stratified sample from the official test set, ensuring equal representation of chart types (e.g., bar, line, pie). This measures performance on data from the same distribution as the training set.
- **Near-Domain (ChartBench Xu et al. (2023)):** A similarly stratified sample from Chart-Bench. This benchmark, while out-of-domain (OOD), shares structural and stylistic similarities with ChartX, testing for near-transfer capabilities.
- **Far-Domain (ChartQA Masry et al. (2022)):** A random sample from the human-annotated portion of the test set. These examples often require deeper, qualitative reasoning, posing a rigorous challenge to purely quantitative methods.
- **Far-Domain (CharXiv Wang et al. (2024)):** A random sample from CharXiv, which contains "in-the-wild" scientific charts with significant visual complexity and stylistic diversity. This serves as a stress test for generalization.

This multi-faceted evaluation was designed to reveal how a strategy optimized for clean, structured data would perform when confronted with the ambiguities and complexities of real-world visualizations.

**Detailed Analysis.**   The results in Table 8 reveal a sharp dichotomy in generalization performance. The code-based specialist (`SFT, Code-based CoT`) excelled on structured data, achieving an impressive **71.6%** on ChartX. However, this rigid strategy proved brittle when generalized, with accuracy plummeting on complex charts like CharXiv to just **18.4%**. This shows how reasoning patterns effective for simple charts become detrimental when misapplied. Furthermore, this failure is not a simple matter of competence that can be fixed with more training. Optimizing the policy with reinforcement learning (RL) or maximizing coding skill on a vast dataset (`CPT + RL`) failed to resolve this core conflict.

## B   CASE STUDY: FAILURE DUE TO NUMERICAL HALLUCINATION

As discussed in Section 3, a critical failure mode for rigid, code-based strategies is *numerical hallucination*. This occurs when the model incorrectly perceives the visual information in a chart

**Table 8:** Detailed performance of "One-Size-Fits-All" Strategies. This table provides the full numerical data visualized in Figure 2 in the main text. All models are fine-tuned (SFT or RL) on the ChartX validation set. The CPT model first undergoes continued pre-training on Chart2Code-160k Zhao et al. (2025b) to enhance its core chart-to-code ability. Despite optimization, no single strategy excels across all benchmarks, revealing a fundamental performance trade-off.

| Prompt Strategy | Training Method | ChartX | ChartBench | ChartQA | CharXiv | Average |
|---|---|---|---|---|---|---|
| | Base Model | 59.8 | 51.6 | 80.4 | 38.2 | 58.7 |
| Standard CoT | SFT on ChartX | 69.8 | 56.2 | 72.0 | 37.2 | 58.8 |
| | RL on ChartX | 63.0 | 53.0 | **81.6** | **39.4** | 59.3 |
| | Base Model | 62.6 | 53.4 | 74.8 | 29.8 | 55.2 |
| Code-as-Thought | SFT on ChartX | **71.6** | **56.8** | 68.2 | 18.4 | 53.8 |
| | RL on ChartX | 66.6 | 55.8 | 78.0 | 37.0 | **59.4** |
| | CPT + RL on ChartX | 69.2 | 54.0 | 68.6 | 32.0 | 56.0 |

and generates flawed code based on this misperception. The model then proceeds to execute its own flawed logic, leading to an answer that is logically consistent with its internal (wrong) representation but factually incorrect.

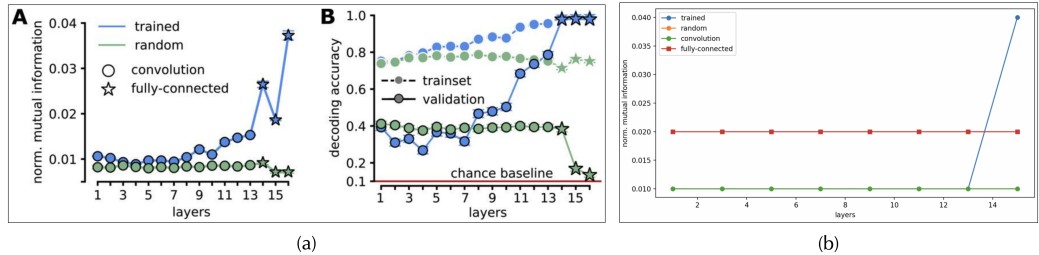

[**Question**]: How many number of datapoints for convolution in the figure A?
[**Ground Truth**]: 26
[**Answer**]: The code is ```python\n{code}\n```. The total number of datapoints for convolution in the figure A is 8. \box{8}

**Figure 7: Failure of a Rigid Code-Based Strategy on a CharXiv Example.** The model is tasked with analyzing the original chart (a) from the CharXiv dataset. It generates Python code (indicated in the red box) to extract the data, but this code hallucinates an incorrect data structure. Chart (b) is the visualization produced by *executing the model's flawed code*. The model then faithfully reasons over its own erroneous chart (b) to arrive at the answer '8', a stark deviation from the ground truth of 26. This case exemplifies how a rigid code-based approach can fail by building logical conclusions on a foundation of numerical hallucination.

## C    ANNOTATION FRAMEWORK FOR VISUAL PROGRAMMABILITY

To train a model capable of recognizing Visual Programmability, we developed a rigorous annotation framework grounded in expert human judgment. We chose this approach because the boundary between visual and symbolic representation is fundamentally cognitive; it involves nuanced, tacit knowledge that is difficult to capture with purely algorithmic rules.

### C.1    GUIDING PRINCIPLE

We built our methodology around a single, functional question for annotators: *"Does a code-based representation preserve the essential information required to correctly answer this question?"* This principle ensures that every label is context-aware, reflecting how the task depends on both the chart's properties and the user's specific query.

### C.2    ASSESSMENT CRITERIA

Annotators evaluated each chart-question pair using a two-step assessment designed to mirror the decision-making process we want our model to learn.

- **Primary Assessment: Information Preservation.** The core question was whether the chart's essential information could be faithfully translated into code. Annotators considered if the underlying data could be reliably extracted from visual elements (e.g., bar heights, point positions) and if this programmatic format would retain everything needed to answer the question. If critical information was lost in this translation—such as the meaning conveyed by complex annotations, visual metaphors, or specific color gradients—the instance was marked as having low programmability for that task.

- **Secondary Assessment: Reconstruction Feasibility.** As a practical test, annotators performed a "mental compilation." They envisioned how the chart might be programmatically recreated using a standard plotting library like Matplotlib. If key visual elements or context could not be captured in this hypothetical reconstruction, it served as a strong signal for low programmability.

### C.3 Annotation Process and Quality Control

To ensure the quality and consistency of our dataset, we followed a structured process.

- **Binary Categorization.** For practical model training, we classified each instance into one of two categories: **high programmability** (suitable for code-based reasoning) or **low programmability** (requires direct visual reasoning). This binary choice frames the model's learning objective as a clear, decisive action.

- **Systematic Guidelines.** All annotations were guided by a detailed rulebook. In ambiguous or boundary cases, annotators were instructed to be conservative, prioritizing the integrity of the visual information over forcing a programmatic representation.

- **Quality Assurance.** We regularly reviewed batches of annotated samples to ensure adherence to our guidelines. This iterative validation process helped maintain high levels of consistency and quality throughout the dataset.

By grounding our dataset in this human-centric process, we provide our model with a supervisory signal that reflects the nuances of human cognition. This enables it to learn a flexible, adaptive policy for chart understanding that moves beyond the limitations of rigid, rule-based systems.

## D Experimental Setup Details

**Training Data.** Our training is based on the ChartMimic Yang et al. (2024) dataset, which contains 4,800 diverse chart–code pairs without QA. To support adaptive learning, we expand this dataset by generating new question–answer pairs with Gemini-2.5-Flash Comanici et al. (2025), using the prompts in Appendix F. This yields a balanced training set that includes charts well-suited for code-based reasoning as well as those demanding direct visual interpretation.

**Evaluation Benchmarks.** We evaluate across four benchmarks chosen to span a wide range of Visual Programmability:

- **ChartX** Xia et al. (2024): high-programmability charts (1,152 structured plots), ideal for testing code-based reasoning.

- **ChartBench (NQA)** Xu et al. (2023): numerical reasoning where data points are not explicitly labeled; we use 2,000 NQA samples to test programmatic extraction from visual cues.

- **ChartQA** Masry et al. (2022): 2,396 real-world charts with human/template questions, covering basic retrieval to multi-step reasoning.

- **CharXiv** Wang et al. (2024): low-programmability, in-the-wild scientific charts (1,323 plots) stressing robustness when code is unreliable.

**Training Details.** We initialize Qwen2.5-VL-7B and train it with EasyR1 Zheng et al. (2025) using GRPO (objective in Appendix G), guided by the multi-component reward in Eq. 2. After validation tuning, weights are set to: answer accuracy $w_{\text{acc}}=0.8$, decision appropriateness $w_{\text{decision}}=0.3$, data

fidelity $w_{\text{data}}$=0.15, and format compliance $w_{\text{format}}$=0.05. All prompts appear in Appendix F. A complete list of hyperparameters and implementation specifics is provided in Appendix J.

# E   DATA ACCURACY REWARD IMPLEMENTATION

The Data Accuracy Reward ($r_{\text{data}}$) is a critical component for ensuring that the model's generated code is not only syntactically correct but also faithfully extracts the data from the chart. This reward is calculated by comparing the DataFrame generated by the model's code against a ground-truth CSV. The full process is detailed in Algorithm 1.

---

**Algorithm 1** Data Accuracy Reward Computation

---

**Require:** Generated code $\mathbf{c}_{\text{pred}}$, Ground truth CSV $\mathbf{csv}_{\text{gt}}$
**Ensure:** Data accuracy reward $r_{\text{data}}$
 1: Extract DataFrame construction code from $\mathbf{c}_{\text{pred}}$ using AST parsing
 2: $\text{DF}_{\text{pred}} \leftarrow$ CONSTRUCTDATAFRAME(extracted_data)
 3: $\text{DF}_{\text{gt}} \leftarrow$ PARSECSV($\mathbf{csv}_{\text{gt}}$)
 4: **if** $\text{DF}_{\text{pred}}$ is None or $\text{DF}_{\text{gt}}$ is None **then**
 5:     **return** 0.0
 6: **end if**
 7:                                                                ▷ Column Completeness Score
 8: matched_cols $\leftarrow$ 0
 9: **for** each column $c_{\text{ref}}$ in $\text{DF}_{\text{gt}}$ **do**
10:     $c_{\text{ref}}^{\text{norm}} \leftarrow$ NORMALIZE($c_{\text{ref}}$)                          ▷ Remove spaces, lowercase
11:     best_match $\leftarrow$ FUZZYMATCH($c_{\text{ref}}^{\text{norm}}$, $\text{DF}_{\text{pred}}$.columns)
12:     **if** match_score $> 50$ **then**
13:         matched_cols $\leftarrow$ matched_cols + 1
14:     **end if**
15: **end for**
16: $r_{\text{col}} \leftarrow$ matched_cols / len($\text{DF}_{\text{gt}}$.columns)
17:                                                                ▷ Row Completeness Score
18: $r_{\text{row}} \leftarrow \mathbb{1}[\text{len}(\text{DF}_{\text{pred}}) = \text{len}(\text{DF}_{\text{gt}})]$
19:                                                                ▷ Value Accuracy Score
20: total_accuracy $\leftarrow$ 0
21: compared_cols $\leftarrow$ 0
22: **for** each matched column pair $(c_{\text{pred}}, c_{\text{gt}})$ **do**
23:     correct_values $\leftarrow$ 0
24:     **for** each row $i$ in min(len($\text{DF}_{\text{pred}}$), len($\text{DF}_{\text{gt}}$)) **do**
25:         **if** COMPAREVALUES($\text{DF}_{\text{pred}}[c_{\text{pred}}][i]$, $\text{DF}_{\text{gt}}[c_{\text{gt}}][i]$) **then**
26:             correct_values $\leftarrow$ correct_values + 1
27:         **end if**
28:     **end for**
29:     col_accuracy $\leftarrow$ correct_values / num_comparisons
30:     total_accuracy $\leftarrow$ total_accuracy + col_accuracy
31:     compared_cols $\leftarrow$ compared_cols + 1
32: **end for**
33: $r_{\text{values}} \leftarrow$ total_accuracy / compared_cols
34:                                                                ▷ Combined Data Accuracy Score
35: $r_{\text{data}} \leftarrow 0.2 \cdot r_{\text{col}} + 0.1 \cdot r_{\text{row}} + 0.7 \cdot r_{\text{values}}$
36: **return** $r_{\text{data}}$

---

The COMPAREVALUES function is designed to be robust. For numerical values, it uses a relative tolerance of $10^{-2}$ to handle minor extraction or floating-point discrepancies. For textual values, it performs case-insensitive, normalized string matching. It also correctly handles NaN values, returning true only if both values are NaN.

## F    DETAILED PROMPT SPECIFICATIONS

This section details the key prompts used for data generation and model training. The prompt for generating synthetic question-answer pairs is presented in Prompt F. The baseline prompts for direct Chain-of-Thought and mandatory code-based reasoning are shown in Prompt F and Prompt F, respectively. Finally, the master prompt that guides our adaptive model to learn strategy selection is detailed in Prompt F.

---

**Prompt for Synthetic Question-Answer Pair Generation**

```
You are a specialized generator of chart comprehension
questions.Using (i) a chart graphic and (ii) the Python
code that creates it, formulate **one** question with
its correct answer.
### Guidelines
1. Answers must come from chart observation and code
understanding
2. Provide exactly one brief, precise response with
**no additional details**
3. Avoid multiple choice, yes/no, or lengthy
descriptive formats
4. Emphasize questions requiring data interpretation expertise
5. Keep answers short (numbers, percentages, names, dates,
or brief terms)
### Question Categories
#### **Numerical Operations**
- **Counting Tasks**: Enumerate items, groups, or elements
with properties
- **Basic Mathematics**: Addition, subtraction, multiplication,
division
- **Descriptive Statistics**: Average, median, mode, range,
maximum, minimum
- **Ratio Analysis**: Proportional relationships
between categories
- **Conditional Analysis**: Elements meeting specific
requirements
- **Multi-step Problems**: Combined computational
operations
#### **Object Recognition**
- **Ranking Identification**: Highest or lowest performing
entities
- **Peak Value Location**: Items with extreme measurements
- **Group Classification**: Category membership identification
- **Time-based Analysis**: Performance identification across
periods
- **Benchmark Comparison**: Items relative to specific
standards
#### **Comparison Tasks**
- **Head-to-head Analysis**: Direct comparison between entities
- **Position Ranking**: Order determination in sequences
- **Variation Analysis**: Largest differences between items
#### **Temporal Analysis**
- **Trend Identification**: Increase/decrease periods
- **Change Detection**: Significant transition moments
- **Pattern Analysis**: Cyclical or seasonal behaviors
### Answer Types
- **Numeric**: '92', '4.2', '17%'
- **Monetary**: '$2,100', '£1,400'
```

```
- **Names**: 'Samsung', 'India', '2022'
- **Categories**: 'Transportation', 'Media'
- **Time**: 'August', 'Q4', '2018'
- **Ratios**: '3:5', '1.7'
### Output Format
```json
{{"question": "Question text here", "answer": "Short answer"}}
```
### Task
**Chart Image**: <image>
**Python Code**: {python_files}
Develop one JSON question-answer pair.
```

**Baseline Chain-of-Thought (CoT) Prompt**

```
Carefully examine this chart. Based on your observations,
answer the question. Let's reason step by step,
then put your final answer under format \\boxed{}.
```

**Code-based Chain-of-Thought (Code-CoT) Prompt**

```
You must carefully examine the chart and the question. First
redraw the image using Python code. This code should aim to
focus on data accuracy and basic chart type representation.
The code must be runnable. Before any plotting, import pandas
and construct one 'pandas.DataFrame' named 'chart_data' that
contains all raw numerical data you will use. The DataFrame
must include appropriate column names and keep the original
row order. Then describe your step-by-step thought process
and answer the question using a single word or phrase and
put it under format \\boxed{}.
```

**Master Prompt for the Adaptive Reasoning Framework**

```
You are an expert at analyzing charts and answering questions
about them. You have two powerful approaches, with code-based
analysis being your preferred method when applicable.
## Core Principle
Code-based analysis is highly effective and should be your
first choice when charts contain extractable data. Code
provides precision, reproducibility, and often superior
accuracy compared to visual estimation.

## Approach Selection Examples
**Example 1:** Bar chart with clear axis labels and readable
values
- Question: "What's the average value across all bars?"
- Best Choice:  (Perfect for extracting values and
calculating precisely)

**Example 2:** Complex 3D visualization or heavily artistic
infographic
- Question: "What trend does this show?"
- Best Choice: <DIRECT> (Data extraction would be unreliable
here)
```

```
## When to Use Each Approach
### Use  When Charts Are Analyzable:
- **Any Standard Chart**: Bar, line, pie, scatter,
histogram - even if slightly messy
- **Readable Data Points**: If you can see numbers
or estimate from gridlines - **use code!**
- **Clear Structure**: Regular patterns, axes,
legends - perfect for code extraction
- **Questions Needing Precision**: Calculations, comparisons,
trends - code gives exact answers
- **Moderate Complexity**: Don't avoid code just because
extraction takes effort - be brave!

### Use <DIRECT> Only When Code Is Truly Impractical:
- **Extremely Artistic/Stylized**: Heavy design elements
completely obscure data structure
- **No Readable Scale**: Completely missing or unintelligible
axes
- **Pure Qualitative**: Questions only about general patterns,
not specific values
- **Severely Distorted**: 3D effects or perspectives that make
extraction impossible

## **Decision Framework**
**Step 1: Code Preference Check**
- Can I see any numerical data or gridlines? → **TRY **
- Are there clear bars, lines, or data points? → **TRY **
- Would precise calculations help answer this question? →
**TRY **

**Step 2: Only if Step 1 fails**
- Is the chart purely artistic with no extractable structure?
→ Use <DIRECT>
- Is the question purely qualitative? → Use <DIRECT>

## Response Format
**First, make your choice with confidence:**
- For code-assisted analysis: output 
- For direct analysis: output <DIRECT>

### If Using Code-Assisted Analysis ():
**Start with**: 
Then proceed with your analysis using code as helpful. Before
any coding, import pandas and construct one `pandas.DataFrame`
named `chart_data` that contains all raw numerical data you
will use. The DataFrame must include appropriate column names
and keep the original row order. You may:
- Redraw/recreate the chart data for comprehensive analysis
- Use code for calculations, comparisons, or data processing
- Combine visual observations with computational analysis
- Focus on the most relevant chart elements for the question

*Note: Choose the code approach that best fits the chart and
question - full redrawing, partial extraction, or targeted
calculations.*

### If Using Direct Analysis (<DIRECT>):
```

```
**Start with**: <DIRECT>
Then provide your reasoning and analysis in the most effective
way for the question. Consider:
- Key observations and findings from the chart
- Your reasoning process and logical steps
- Relevant patterns or trends you identify

## Final Answer Format
Every response MUST end with \\boxed{your_answer}

Now analyze the given chart and question. Choose your approach
based on the chart's extractability and the question's
requirements.
```

## G  GRPO OBJECTIVE AND UPDATE DETAILS

We employ Group Relative Policy Optimization (GRPO) Shao et al. (2024), a policy-gradient method that leverages group-normalized advantages and PPO-style clipping. For each training instance $x$, we sample a group of $G$ responses $\{r_i\}_{i=1}^{G}$ from the previous policy $\pi_{\text{old}}$, evaluate them with our reward $R$, and update the current policy $\pi_\theta$ by maximizing:

$$J_{\text{GRPO}}(\theta) = \mathbb{E}\left[\sum_{i=1}^{G} \min\left(\frac{\pi_\theta(r_i \mid x)}{\pi_{\text{old}}(r_i \mid x)} A_i, \ \text{clip}\left(\frac{\pi_\theta(r_i \mid x)}{\pi_{\text{old}}(r_i \mid x)}, \ 1-\epsilon, \ 1+\epsilon\right) A_i\right) - \beta D_{\text{KL}}(\pi_\theta \| \pi_{\text{ref}})\right],$$

(3)

where the group-normalized advantage is

$$A_i = \frac{R(r_i, \phi_i) - \text{mean}(\{R(\xi, \phi)\}_{\xi=1}^{G})}{\text{std}(\{R(\xi, \phi)\}_{\xi=1}^{G})}.$$

(4)

Here, $\epsilon$ controls the clipping range, and $\beta$ weights the KL penalty against a reference policy $\pi_{\text{ref}}$. In our final configuration, we set $\beta=0$ (KL-free), focusing purely on the group-relative signal.

**Practical Notes.**

- **Sampling.** For each $x$, draw $G$ responses from $\pi_{\text{old}}$; compute rewards and $A_i$ using the group statistics.
- **Clipped update.** Define ratio $\rho_i = \pi_\theta(r_i|x)/\pi_{\text{old}}(r_i|x)$ and apply $\min(\rho_i A_i, \text{clip}(\rho_i, 1-\epsilon, 1+\epsilon) A_i)$.
- **No-KL setup.** Use $\beta=0$; we found this configuration works well with verifiable rewards.

## H  BROADER IMPLICATIONS AND FUTURE DIRECTIONS

Our work on adaptive chart reasoning, while focused on a specific domain, offers insights into a broader challenge in artificial intelligence: developing systems that can flexibly navigate between different problem-solving strategies. Just as humans alternate between rapid, intuitive pattern recognition and slower, deliberate symbolic reasoning Kahneman (2011), future AI systems must master not only individual skills but also the meta-level ability to select the right tool for the job.

**From Modality Fusion to Method Fusion.**  Much of the research in multimodal AI has centered on *modality fusion*—the effective combination of information from different sensory channels. Our framework points towards a complementary and perhaps equally important paradigm: *method fusion*. This refers to the ability to select and combine different reasoning strategies (e.g., visual-perceptual vs. symbolic-programmatic) even when operating within a single modality. The challenge is not only to perceive the world through multiple senses but to think about it through multiple "lenses," fluidly shifting between holistic, pattern-based analysis and precise, step-by-step decomposition as the problem demands.

**Competence Awareness as a Foundational Capability.** A key takeaway from our research is that models can be trained to recognize the boundaries of their own competence with respect to specific methods. This nascent form of meta-cognitive awareness—knowing not just *how* to solve a problem, but knowing *which* of its available methods is most likely to succeed—is a fundamental prerequisite for robust and reliable AI. We foresee that future general-purpose systems will need to develop richer internal models of their own capabilities, enabling them to make more dependable strategy selections when faced with novel tasks.

**Limitations and Key Future Directions.** While our adaptive framework represents a significant step, its current limitations highlight critical areas for future research that build directly upon our findings.

- **Granular and Hybrid Reasoning:** The current decision-making process is a binary choice between "code" and "direct" reasoning. This could be extended to a more granular **hybrid model**, where code is used for reliable data extraction while visual reasoning concurrently interprets qualitative patterns from the same chart. Furthermore, assessing programmability at the chart-level is coarse; future models could learn to perform **region-based assessment**, applying different strategies to different parts of a single complex figure.

- **Expanding the Vocabulary of Formal Reasoning:** Our model's "Code as Thought" process is currently centered on data analysis logic. A natural evolution is to expand the scope of this *native* formal reasoning. Instead of orchestrating external tools, future work could enrich the model's internal symbolic language to encompass other formalisms, such as the logic of signal processing for time-series charts or graph-theoretic principles for network diagrams. This would extend the reach of the model's innate symbolic capabilities, allowing it to tackle a wider range of problems programmatically without breaking the "native reasoning" paradigm.

- **Self-Supervised Policy Learning:** A key challenge is reducing the reliance on annotated training data for programmability. A promising direction is developing **self-supervised methods** where the model learns the decision boundary by correlating its choice of strategy with final task success. This would effectively teach the model to recognize the reliable application range of its own internal methods without requiring explicit human-provided labels.

**Towards Dynamic Strategy Composition.** Looking further ahead, our current framework makes a discrete selection between predefined strategies. A significant extension would be for future systems to dynamically *compose* novel strategies from a set of primitive cognitive operations. For instance, when analyzing a complex visualization, an advanced system might synthesize a hybrid approach on the fly: invoking its internal graph-based logic for structural analysis while using its time-series forecasting logic for temporal patterns. This compositional flexibility, guided by a learned meta-policy, would represent a significant leap towards more human-like adaptability.

**The Path Forward.** The journey from narrow tools to general intelligence will likely require architectural innovations that foster cognitive flexibility. Our adaptive framework, though applied to chart understanding, provides a concrete instantiation of these principles. By teaching a model to recognize when formal reasoning is a powerful asset versus a brittle liability, we take a tangible step toward systems that reason not just powerfully, but appropriately. The ultimate goal is not to build models that always default to their most complex methods, but ones that can gracefully match their computational effort and reasoning style to the structure of the problem at hand—a hallmark of true intelligence.

## I  DETAILED CASE STUDIES

This section provides the full visualizations and detailed model outputs for the qualitative analysis. Each case includes the figure, the task details, and the verbatim model outputs from both a baseline and our adaptive model.

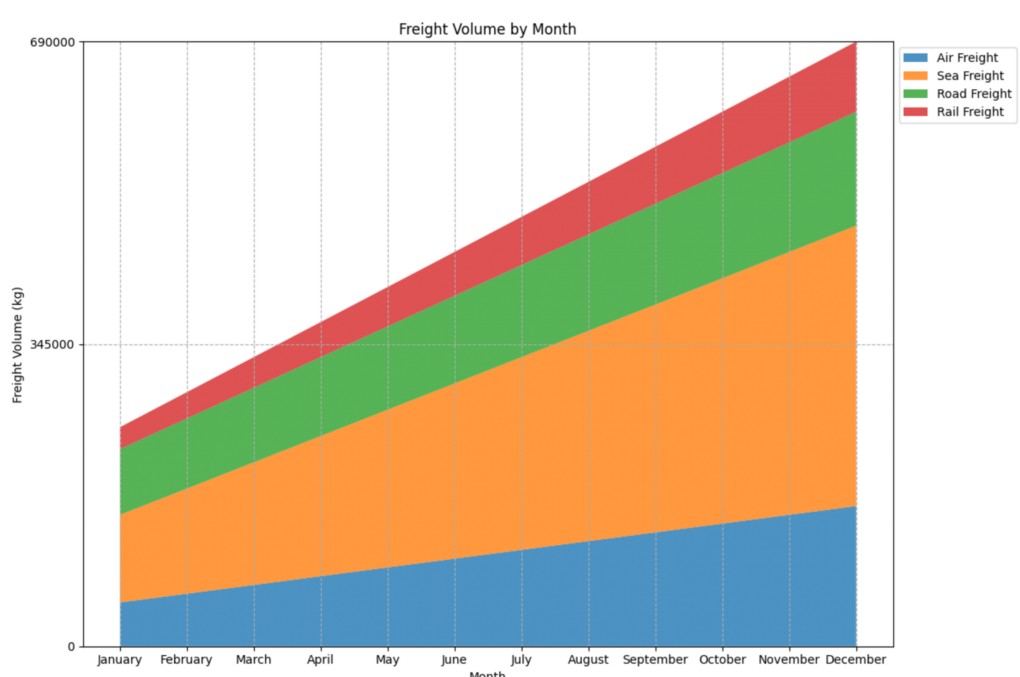

[**Question**] How much did the sea freight volume increase from January to December?
[**ground_truth**] 220000 kg

[**CoT**]
To determine the increase in sea freight volume from January to December, we need to follow these steps:\n1. Identify the sea freight volume in January.\n2. Identify the sea freight volume in December. 3. Subtract the January volume from the December volume to find the increase.\n\nLet's start by estimating the sea freight volumes for January and December from the stacked area chart:\n\n- In January, the sea freight volume appears to be around 100,000 kg.\n\n- In December, the sea freight volume appears to be around 250,000 kg.\nNow, we calculate the increase:\n\n\[\n\text{Increase} = \text{December volume} - \text{January volume} = 250,000 \, \text{kg} - 100,000 \, \text{kg} = 150,000 \, \text{kg}\n\]\n\nThus, the increase in sea freight volume from January to December is \boxed{150000}.

[**Adaptive(ours)**]
\n\nFirst, let's reconstruct the chart data into a pandas DataFrame for analysis:\n\n```python\nimport pandas as pd\n\ndata = {\n'Month': ['January', 'February', 'March', 'April', 'May', 'June', 'July', 'August', 'September', 'October', 'November', 'December'],\n'Air Freight': [100000, 120000, 140000, 160000, 180000, 200000, 220000, 240000, 260000, 280000, 300000, 320000],\n'Sea Freight': [150000, 170000, 190000, 210000, 230000, 250000, 270000, 290000, 310000, 330000, 350000, 370000],\n'Road Freight': [70000, 80000, 90000, 100000, 110000, 120000, 130000, 140000, 150000, 160000, 170000, 180000],\n'Rail Freight': [20000, 25000, 30000, 35000, 40000, 45000, 50000, 55000, 60000, 65000, 70000, 75000],\n}\n\nchart_data = pd.DataFrame(data)\n```\n\nNow, we will analyze the sea freight volume increase from January to December.\n\nFrom the chart_data:\n- Sea Freight in January = 150000 kg\n- Sea Freight in December = 370000 kg\n\nThe increase in sea freight volume from January to December is:\n```python\nincrease = chart_data.loc[chart_data['Month'] == 'December', 'Sea Freight'].values[0] - chart_data.loc[chart_data['Month'] == 'January', 'Sea Freight'].values[0]\nincrease\n```\n\n\n\n<DIRECT>\nThe sea freight volume in January is 150000 kg and in December it is 370000 kg. The increase in sea freight volume from January to December is:\n\[ 370000 - 150000 = 220000 \text{ kg} \]\n\n\\boxed{220000}\n

**Figure 8:** Case Study 1: High-Programmability Chart from ChartX. On a standard stacked area chart from ChartX that required precise calculation, our adaptive model correctly chose the  path, extracting exact data and computing the correct answer. In contrast, a fixed 'Standard CoT' model relied on visual estimation and failed. This shows the model's ability to leverage code for precision.

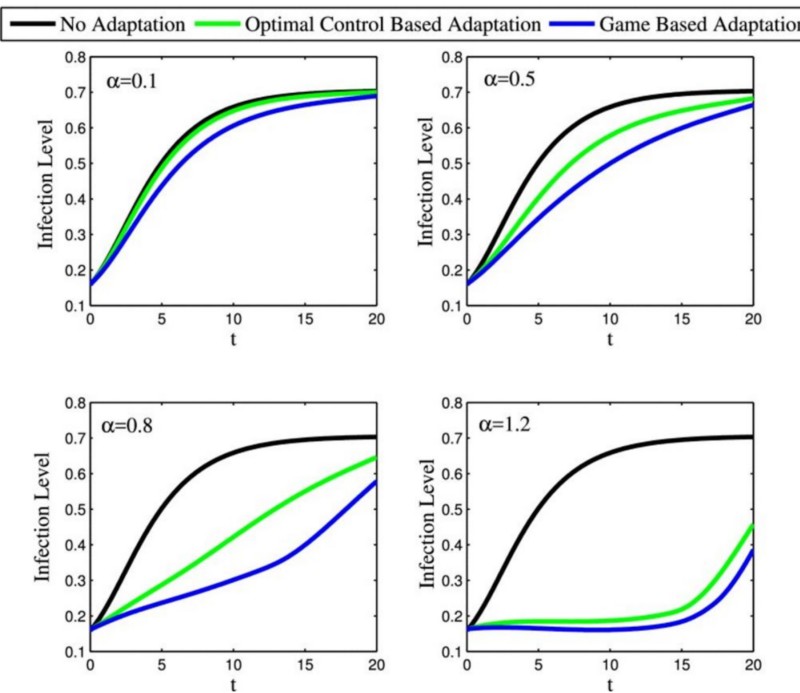

[**Question**] What is the value of the α of the subplot where the Optimal Control Based Adaptation and Game Based Adaptation has most difference in Infection Level at t = 15?
[**ground_truth**] 0.8

[**Code-CoT**]
First, let's create a DataFrame `chart_data` with the necessary data:\n\n```python\nimport pandas as pd\n\nchart_data = pd.DataFrame({\n'alpha': [0.1, 0.5, 0.8, 1.2],\n'no_adaptation': [0.7, 0.7, 0.7, 0.7],\n'optimal_control': [0.7, 0.7, 0.65, 0.45],\n'game_based': [0.7, 0.65, 0.55, 0.4]\n})\n```\n\nNow, let's plot the data using Matplotlib. Here is the code to create the plot:\n\n```python\nimport matplotlib.pyplot as plt\n\nfig, axs = plt.subplots(2, 2, figsize=(10, 8))\nfor ax, alpha, no_adaptation, optimal_control, game_based in zip(axs.flatten(), chart_data['alpha'], chart_data['no_adaptation'], chart_data['optimal_control'], chart_data['game_based']):\nax.plot(chart_data['t'], no_adaptation, label='No Adaptation', color='black')\nax.plot(chart_data['t'], optimal_control, label='Optimal Control Based Adaptation', color='green')\nax.plot(chart_data['t'], game_based, label='Game Based Adaptation', color='blue')\nax.set_xlabel('t')\nax.set_ylabel('Infection Level')\nax.set_title(f'α={alpha}')\nax.set_xlim(0, 20)\nax.set_ylim(0.1, 0.8)\nax.legend()\nplt.show()\n```\n\nNow, to answer the question: We need to find the value of α where the difference between Optimal Control Based Adaptation and Game Based Adaptation is the most significant at t = 15.\n\nFrom the plots:\n- At α = 0.1, the difference is minimal.\n- At α = 0.5, the difference is moderate.\n- At α = 0.8, the difference is significant.\n- At α = 1.2, the difference is the most significant.\n\nThus, the value of α where the difference is the most significant at t = 15 is:\n\n\\boxed{1.2}

[**Adaptive(ours)**]
The question asks for the value of α where the difference between Optimal Control Based Adaptation and Game Based Adaptation is most significant at t = 15. From the visual inspection of the subplots, it appears that this difference is most pronounced in the subplot where α = 0.8.\n\n<DIRECT>\nAt t = 15, the infection level for α = 0.8 is significantly different between the Optimal Control Based Adaptation (green line) and the Game Based Adaptation (blue line). The green line (Optimal Control Based Adaptation) is higher than the blue line (Game Based Adaptation) at this point.\n\n\\boxed{0.8}\n

**Figure 9:** Case Study 2: Low-Programmability Chart from CharXiv. Faced with a complex scientific plot from CharXiv requiring qualitative comparison, a fixed code-based model failed by hallucinating a data table. Our adaptive model, however, correctly identified the task's low programmability and chose the <DIRECT> path. It performed a robust visual comparison, leading to the correct answer and demonstrating its critical skill in avoiding tools when they are unsuitable.

## J  IMPLEMENTATION AND HYPERPARAMETER DETAILS

Our model was trained using the configuration and hyperparameters summarized in Table 9. We used the EasyR1 Zheng et al. (2025) framework for our reinforcement learning implementation. The base model, Qwen2.5-VL-7B, was trained for 200 episodes. The vision tower of the model remained frozen during training to preserve its pre-trained perceptual capabilities.

**Table 9:** Training Configuration Details

| Configuration | Value |
|---|---|
| *Model Configuration* | |
| Base Model | Qwen2.5-VL-7B |
| Vision Tower | Frozen |
| Precision | BFloat16 |
| Max Prompt Length | 5,120 tokens |
| Max Response Length | 3,072 tokens |
| *Data Configuration* | |
| Seed | 42 |
| Shuffle | True |
| Filter Overlong Prompts | True |
| *Training Hyperparameters* | |
| Algorithm | GRPO (without KL penalty) |
| Learning Rate | $1.0 \times 10^{-6}$ |
| Optimizer | AdamW (BF16 variant) |
| Global Batch Size | 64 |
| Rollout Batch Size | 256 |
| Micro Batch Size (Update) | 4 |
| Micro Batch Size (Experience) | 16 |
| Training Episodes | 4 |
| Gradient Clipping | 1.0 |
| *Rollout Configuration* | |
| Number of Rollouts ($n$) | 5 |
| Temperature | 1.0 |
| Top-p | 0.99 |
| *Infrastructure* | |
| GPUs | $8 \times$ NVIDIA H800 |
| Tensor Parallelism | 1 |
| FSDP | Enabled |
| CPU Offloading | Disabled |
| Gradient Checkpointing | Enabled |
| *Validation* | |
| Validation Batch Size | 512 |
| Validation Frequency | Every 5 episodes |
| Validation before Training | Yes |

