# OpenReview forum: "Visual Programmability: A Guide for Code-as-Thought in Chart Understanding"
_ICLR.cc/2026/Conference — ICLR 2026 Conference Withdrawn Submission_

### Official Review · Reviewer_t3mh · 2025-10-18

**Soundness:** 2
**Presentation:** 2
**Contribution:** 2
**Rating:** 2
**Confidence:** 4

**Summary:**

This paper introduced Visual Programmability, a RL framework for VLMs to determine if a chart-question pair is better to solve with code or without code. The motivation of this work is from that using code to solve a task or not is task-dependent. Some tasks can benefit from using code, while other not. The proposed methods achieves better performance on five benchmarks compared to other similiar-scale models. Extensive ablation studies are performed to support the effectiveness of the proposed method.

**Strengths:**

- This paper captures a keen observation that how and when to use tools is crucial for VLMs, and I believe this is an important research problem.
- This paper also performed extensive experimental results to show the effectiveness of the proposed method.

**Weaknesses:**

- This paper only focuses chart-based QA, which is quite limited, other mathematical-type visual tasks, or other general visual tasks may benefit from using tools as well.
- The proposed training method relies on the pre-annotated visual programmability, which is less described in the paper. However, I think this is the core of the algorithm and should be highlighted.
- This paper lacks technical novelty, for typical tool-use training, model will learn when and how to call tools adaptively. How this framework if different to the conventional tool-use training?

**Questions:**

- I am a bit confused by the CaT, is the python tool called when model choose to use CaT? From Figure 3, this seems to be the case, but the paper dose not discuss how tools are used during training and inference.
- How the decision reward is computed exactly?

---

### Official Review · Reviewer_uXzj · 2025-10-22

**Soundness:** 3
**Presentation:** 2
**Contribution:** 2
**Rating:** 4
**Confidence:** 4

**Summary:**

This paper introduces the concept of Visual Programmability for chart reasoning with vision-language models (VLMs). The central idea is that not all chart-question pairs benefit equally from executable code reasoning; some are better solved through direct visual understanding. The proposed framework equips an MLLM with the ability to first decide between two strategies and then to follow the corresponding reasoning path. Experiments show consistent improvements compared to single-strategy baselines.

**Strengths:**

The paper is well motivated and tackles an important problem in chart reasoning, where tasks vary in how much they benefit from executable code versus direct perception. The idea of learning a strategy token that dynamically chooses between reasoning modes is simple yet effective. The probabilistic formulation is sound, and the reinforcement learning setup with multiple reward components is sensible, especially the decision reward that prevents mode collapse. Experiments cover multiple datasets and show consistent improvement.

**Weaknesses:**

I think the main limitation lies in the transparency and robustness of the programmability supervision. The decision reward relies on pre-labeled “Visual Programmability” indicators, but the paper does not report how these labels were produced, e.g., who annotated them, the inter-annotator agreement, or how disagreement was resolved. Appendix C briefly outlines the labeling process but lacks quantitative metrics. To make this part credible, the authors should provide (a) proportions of high/low programmability samples per dataset, (b) number of annotators, (c) Inter-annotator agreement measures such as Cohen’s kappa or Krippendorff’s alpha, and (d) several examples of disagreement cases with final adjudication rules.


The reward design, while intuitively reasonable, lacks sensitivity analysis. Appendix D lists default weights $w_{acc}=0.8$, $w_{decision}=0.3$, $w_{data}=0.15$, and $w_{format}=0.05$, but there is no experiment showing how results vary with these weights. Since multi-objective RL is often unstable, a 2D sweep (e.g., $w_{decision}$ in {0, 0.15, 0.3, 0.6} and $w_{data}$ in {0, 0.05, 0.15, 0.3}) would clarify robustness. The authors should report average accuracy, code usage rate, and variance over three seeds, ideally with confidence intervals. Adding this to Table 5 or as an appendix plot would improve the paper’s rigor.

The fairness of baseline comparison is another issue. Table 3 mixes locally re-run baselines and results copied from prior papers but does not clearly indicate which is which. This weakens the credibility of cross-paper performance claims. It is essential to mark externally sourced results with an asterisk and, for two or three strong baselines, re-run them under the same setup (identical image resolution, parsing template, and prompt format).

The description of the GRPO configuration in Sec. 4.3.1 is somewhat vague. It states that grouped sampling ($G$ samples per prompt) and a “KL-free” variant with $\beta=0$ are used, but key hyperparameters, such as group size $G$, clipping thresholds, temperature, top-p, and advantage normalization method, are missing. These details affect stability and reproducibility.

The data-accuracy reward ($r_{data}$) is innovative but under-evaluated. Algorithm 1 describes fuzzy matching of column names and numerical tolerance (0.01 relative error), but there is no analysis of its precision and recall under adversarial conditions such as column swaps, scaling by units, or noisy text labels. A robustness test with synthetic perturbations would reveal whether $r_{data}$ over-penalizes or under-penalizes outputs.


Another limitation is the hard routing between strategies. Once $P(s|I,Q)$ picks one branch, the model may not recover from a wrong choice. In ambiguous cases, this can produce catastrophic errors, right? The authors could explore a soft routing variant: if the confidence $P(s|I,Q)$ is below a threshold, execute both branches in parallel and select the final answer based on a heuristic such as data-accuracy reward. Even a small-scale experiment on 500 examples could show whether this improves stability.

The dataset generation procedure (Sec. 5.1) mentions that synthetic QA pairs were created using Gemini-2.5-Flash. However, there is no description of the prompt template, filtering process, or deduplication against evaluation sets.

Finally, scale analysis across model sizes (Table 4) needs a clearer explanation. The 3B variant performs worse than the fixed Code-CoT model, yet the 32B model outperforms it. The paper attributes this to model capacity limitations, but no supporting evidence is given. It would be more convincing to test a 3B model with teacher-forced strategy supervision or larger group sampling ($G$), to see whether the gap is architectural or optimization-related.

**Questions:**

- Can you report detailed statistics of Visual Programmability labels, including proportions, annotator counts, and inter-annotator agreement?
- What is the exact configuration of GRPO (group size, ratio clipping, temperature, top-p) and the variance across random seeds?
- Have you tested the robustness of $r_{data}$ under perturbations like column reordering, unit rescaling, or injected noise?
- Did you try soft or hybrid routing (executing both strategies when confidence is low) on any subset? If so, what was the computational overhead and accuracy change?

---

### Official Review · Reviewer_UFCV · 2025-10-29

**Soundness:** 3
**Presentation:** 3
**Contribution:** 2
**Rating:** 4
**Confidence:** 4

**Summary:**

The paper proposes Visual Programmability, a property that helps VLMs decide whether a chart-question pair should be solved with code or direct visual reasoning. The authors design an adaptive framework where the model first predicts if a task is “programmable” and then follows either a Code-as-Thought path or a visual reasoning path. This policy is trained through RL with a dual-reward setup—one for factual accuracy and another for making good strategic decisions.

**Strengths:**

-	The paper is well-written, presenting a clear and logical progression of ideas, supported by effective illustrations.
-	Code is submitted to help the reproducibility.

**Weaknesses:**

- Limited novelty: While the proposed concept is somewhat interesting, the paper lacks innovation in algorithmic design. Apart from the use of reward shaping, most of the training techniques follow existing approaches. The idea of automatically switching between LLM reasoning modes has already appeared in prior work (e.g., AdaCoT [1]). The authors should more clearly state what concrete methodological innovations their work contributes beyond combinations and applications of established methods.

- Insufficient experimental validation: a) Experiments are conducted only on the Qwen2.5-VL model series, which limits the generality of the findings. b) The hyperparameter settings (particularly the weighting of different rewards) appear somewhat ad hoc, and there is a lack of ablation studies to justify these design choices.

- Underwhelming results and missing analysis:
a) As shown in Table 1, the improvements over the baseline are modest (only +1.6pp).
b) In Table 4, the severe performance degradation on the 3B model is not adequately analyzed or explained.

[1] Adacot: https://arxiv.org/pdf/2505.11896

**Questions:**

As a RL task, it would be valuable to evaluate whether the proposed method improves the model’s general-domain capabilities (e.g., on benchmarks such as MMMU). Demonstrating such improvements would strengthen the paper and better highlight the broader impact of the approach.

---

### Official Review · Reviewer_iv3U · 2025-10-31

**Soundness:** 2
**Presentation:** 3
**Contribution:** 2
**Rating:** 4
**Confidence:** 4

**Summary:**

The authors introduce the idea of Visual Programmability to decide whether a visual chart question answering task is better answered with code or with direct perception. They finetune a Qwen-2.5-VL model with GRPO to determine which path to use for a given query and analyze how often code is used in different settings. They show modest benefits on some chart datasets, with ablations for different parts of the reward function used.

**Strengths:**

- The problem is well-motivated: recognizing when to use code-style reasoning is a critical point for code-execution enhanced visual reasoning methods.
- The paper introduces an interesting concept relating to this, Visual Programmability.
- The exploration of how often code is used in different settings is interesting and furthers understanding.
- The annotation of the decision making process for Visual Programmability is a useful contribution.

**Weaknesses:**

- The notion of code-as-thought considered is fairly limited, parsing the chart into structured format and then performing computations. This is in contrast with approaches such as the tool-use approaches discussed as prior work, which can use the VLM itself as one of the tools. The results and idea of visual programmability are presented as properties of the tasks, but depend substantially on how the code-path is defined. It is unclear how much the results would change with different options, like code for operating directly on the image rather than on parsed dataframes. (In the simplest case, visual programs can just be calling the VLM directly.)
- It also seems to depend substantially on how well the parsing step works, but the authors note this in 5.5.2.
- Some reasonable comparisons, such as comparison to in-context examples for whether to use code or images, are missing. For instance, ViperGPT uses in-context examples to distinguish between code that directly calls the VLM on the image and code that performs more complex operations depending on the query. How much is the post-training really helping?
- The empirical benefits of the adaptive setting over the fixed code CoT are unclear (Table 4) even for the higher model size.

**Questions:**

- My biggest question: how well can this be done with in-context examples rather than extensive post-training?
- Why don't the results in Tables 4 (bottom row) and 5 (Full Reward) match?

---

### Note · Authors · 2025-11-19

I have read and agree with the venue's withdrawal policy on behalf of myself and my co-authors.